# Chalkophore-mediated respiratory oxidase flexibility controls *M. tuberculosis* virulence

**John A Buglino[1†], Yaprak Ozakman[1†], Chad E Hatch[2], Anna Benjamin[1], Derek S Tan[2,3], Michael S Glickman[1]***

[1]Immunology Program, Sloan Kettering Institute, New York, United States; [2]Chemical Biology Program, Sloan Kettering Institute, New York, United States; [3]Tri-Institutional Research Program, Memorial Sloan Kettering Cancer Center, New York, United States

**\*For correspondence:**
glickmam@mskcc.org

[†]These authors contributed equally to this work

## eLife Assessment

In this **important** study, the authors advance our understanding of copper uptake by chalkophores and their targeted metalloproteins in *Mycobacterium tuberculosis*. These **convincing** data demonstrate that chalkophore-acquired copper is solely incorporated into the Mtb bcc:aa3 copper-iron respiratory oxidase under low copper conditions, and that chalkophore-mediated protection of the respiratory chain is critical to Mtb virulence. These findings may be leveraged for drug discovery and will be of broad interest to those studying bacterial pathogenesis.

**Abstract** Oxidative phosphorylation has emerged as a critical therapeutic vulnerability of *M. tuberculosis* (*Mtb*). However, it is unknown how intracellular bacterial pathogens such as *Mtb* maintain respiration during infection despite the chemical effectors of host immunity. *Mtb* synthesizes diisonitrile lipopeptides that tightly chelate copper, but the role of these chalkophores in host-pathogen interactions is also unknown. We demonstrate that *M. tuberculosis* chalkophores maintain the function of the heme-copper $bcc:aa_3$ respiratory supercomplex under copper limitation. Chalkophore deficiency impairs *Mtb* survival, respiration to oxygen, and ATP production under copper deprivation in culture, effects that are exacerbated by loss of the heme-dependent Cytochrome BD respiratory oxidase. Our genetic analyses indicate that the maintenance of respiration is the major cellular target of chalkophore-mediated copper acquisition. *M. tuberculosis* lacking chalkophore biosynthesis is attenuated in mice, a phenotype that is also severely exacerbated by loss of the CytBD respiratory oxidase. We find that the host immune pressure that attenuates chalkophore-deficient *Mtb* is independent of adaptive immunity and neutrophils. These data demonstrate that chalkophores counter host-inflicted copper deprivation and highlight a multilayered system by which *M. tuberculosis* maintains respiration during infection.

## Introduction

Bacterial pathogens are subjected to diverse stresses imposed by host immunity and deploy countermeasures to neutralize immune effectors. The ubiquity of metalloenzymes in all bacteria necessitates acquisition of trace metals such as iron, zinc, and manganese, a vulnerability that is exploited by host nutritional immunity, which limits these metals (***Murdoch and Skaar, 2022***). To counter this metal limitation by the host, pathogens deploy diverse high-affinity metal acquisition systems including siderophores for iron and zinc (***Sheldon et al., 2016***; ***Behnsen et al., 2021***; ***Bobrov et al., 2014***), and

**eLife digest** When a bacterium known as *Mycobacterium tuberculosis* infects humans, it can lead to a disease called tuberculosis – one of the leading causes of death from an infectious disease worldwide. The bacteria hide within certain cells in the body so that they are less accessible to the host's immune system.

This approach is so successful that patients typically require long courses of antibiotics lasting many months to eliminate the bacteria. However, the emergence of drug-resistant strains of *M. tuberculosis* means that new methods to target the bacteria are urgently needed.

One possible target for future therapies is a system known as diisonitrile lipopeptide chalkophores, which bind to copper and import the metal into the bacterium from its surroundings. Although certain proteins in bacteria require copper to work properly, the target of the copper acquired by the chalkophores in *M. tuberculosis* has remained unclear.

Here, Buglino, Ozakman et al. used genetic and biochemical approaches to study why diisonitrile lipopeptide chalkophores collect copper in *M. tuberculosis*. The experiments showed that the chalkophores supply copper to proteins with a key role in respiration, the process by which cells make chemical energy needed for many cell activities.

When there was a shortage of copper in their surrounding environment, mutant *M. tuberculosis* cells lacking the chalkophores were less able to produce chemical energy and more likely to die than healthy *M. tuberculosis* cells. This effect was more severe if the cells were also missing an enzyme that enables the bacteria to respire without copper.

Further experiments in mice found that during a tuberculosis infection, the immune system targeted copper-containing proteins in its attempts to kill the bacteria. The mutant *M. tuberculosis* cells were less effective at infecting the mice, suggesting that chalkophores help the bacteria defend themselves against the host immune system.

Taken together, these findings reveal that copper-containing proteins in *M. tuberculosis* are a major target of the immune system. In the future, increasing our understanding of these proteins and identifying drugs that interfere with their activities may lead to new, more effective therapies for tuberculosis.

transporters and metallophores for zinc and manganese (*Murdoch and Skaar, 2022*; *Behnsen et al., 2021*; *Diaz-Ochoa et al., 2016*; *Liu et al., 2012*; *Nairn et al., 2016*). In addition to metal limitation, the host also deploys metals as antimicrobial effectors to kill bacteria (*Sheldon and Skaar, 2019*). Copper and zinc are deposited into the phagosome of infected macrophages as antimicrobials and pathogens employ metal resistance systems such as metal efflux transporters (*Shi and Darwin, 2015*; *Shi et al., 2014*; *Festa et al., 2011*) and metal binding proteins controlled by metal-dependent repressors (*Shi and Darwin, 2015*; *Shi et al., 2014*; *Festa et al., 2011*). Although both metal limitation and metal intoxication limit pathogen growth, in most cases, the essential bacterial metalloenzymes rendered dysfunctional by nutritional immunity are incompletely defined.

*M. tuberculosis* is a successful global pathogen that can survive in both macrophages and neutrophils. *M. tuberculosis* experiences both high copper (*Shi et al., 2014*; *Festa et al., 2011*; *Buglino et al., 2021*; *Rowland and Niederweis, 2012*; *Wolschendorf et al., 2011*; *Wagner et al., 2005*) and high zinc (*Botella et al., 2011*; *Buglino et al., 2021*) in the macrophage phagosome and resists metal toxicity through several mechanisms including efflux and metal chelation. Recent data also indicate that *M. tuberculosis* experiences zinc starvation during infection, possibly imposed by calprotectin in caseum (*Dow et al., 2021*). However, it is unknown whether copper acquisition, and resistance to copper deprivation are part of the virulence program of *M. tuberculosis*. *M. tuberculosis* synthesizes diisonitrile lipopeptide natural products directed by the 5-gene *nrp* operon, present in *M. tuberculosis*, *M. bovis*, and *M. marinum* (*Harris et al., 2018*; *Bhatt et al., 2018*; *Wang et al., 2017*; *Harris et al., 2017*; *Mehdiratta et al., 2022*). The *nrp* operon is induced by copper deprivation and the growth of *M. tuberculosis* lacking diisonitrile lipopeptide biosynthesis is inhibited in copper-limiting conditions and rescued by a synthetic diisonitrile (*Buglino et al., 2022*), establishing diisonitrile lipopeptides as mycobacterial chalkophores. However, the role of diisonitrile chalkophores in *M. tuberculosis* pathogenesis is not understood, including whether copper deprivation is imposed by the host during

infection, and the specific bacterial pathways that require copper supplied by the chalkophores. In this study, we demonstrate that diisonitrile chalkophores supply copper to the heme-copper $bcc:aa_3$ respiratory supercomplex to maintain respiration and ATP production. The heme-copper oxidase of chalkophore-deficient *M. tuberculosis* is compromised by the host during infection, but *M. tuberculosis* compensates with the heme-dependent cytochrome BD (CytBD). *M. tuberculosis* lacking diisonitrile chalkophore biosynthesis and CytBD is severely attenuated, demonstrating that this multilayered system for protecting respiration is a critical virulence function of *M. tuberculosis*.

## Results

### Chalkophore-deficient *M. tuberculosis* upregulates respiratory chain components in response to copper deprivation

To understand the function of diisonitrile chalkophores in *M. tuberculosis*, we examined the transcriptional profile of wild-type (WT) and Δ*nrp* *M. tuberculosis* treated with 10 μM tetrathiomolybdate (TTM), a copper chelator previously shown to inhibit the growth of *M. tuberculosis* lacking chalkophore biosynthesis (*Buglino et al., 2022*). Copper chelation had relatively few effects on gene expression in WT *M. tuberculosis* (*Figure 1A*, *Supplementary file 3*). In contrast, in Δ*nrp* *M. tuberculosis* we observed upregulation of the genes of the chalkophore cluster (other than *nrp* itself) at baseline, consistent with prior data demonstrating autoregulation (*Buglino et al., 2022*). In addition, a cluster of genes was induced by copper deprivation in Δ*nrp* cells, but not WT cells, encoding components of the respiratory chain, including *cydABDC* (encoding components of the heme-dependent oxidase CytBD), *qcrABC* (encoding subunits of the $bcc:aa_3$ heme-copper oxidase), and subunits of ATP synthase (*Figure 1A*). Prior work demonstrated that genetic or pharmacologic disruption of the $bcc:aa_3$ respiratory oxidase, including by Q203, an inhibitor of the QcrB subunit of the $bcc:aa_3$ oxidase, transcriptionally upregulates genes encoding components of the respiratory chain, including *cydAB* and ATP synthase (*Kalia et al., 2017*; *Matsoso et al., 2005*). Treatment of WT and Δ*nrp* with Q203 reproduced the published pattern in WT cells and indicated that diisonitrile chalkophore deficient cells still respond to Q203 (*Figure 1*). To confirm the RNA sequencing results, we quantitated the transcript encoding the *cydA*-encoded subunit of the CytBD oxidase with different concentrations of TTM and observed no induction at any TTM concentration in WT cells, but progressive induction of CydA with escalating TTM concentrations in the Δ*nrp* strain (*Figure 1B*). Longer durations of copper deprivation induced *cydA* in both WT and Δ*nrp*, with higher induction in Δ*nrp* (*Figure 1C*). Escalating concentrations of TTM did not affect WT *M. tuberculosis* growth but caused a graded inhibition of growth of the Δ*nrp* strain in liquid media (*Figure 1D*). These data indicate that copper deprivation in *M. tuberculosis* lacking diisonitrile chalkophore biosynthesis stimulates gene expression that mimics inhibition of the $bcc:aa_3$ respiratory oxidase.

### Chalkophores protect the $bcc:aa_3$ oxidase from copper deprivation

Inhibition of $bcc:aa_3$ oxidase by Q203 (*Kalia et al., 2017*; *Lamprecht et al., 2016*) or other QcrB inhibitors (*Harrison et al., 2019*; *O'Malley et al., 2018*; *Lupien et al., 2020*; *Foo et al., 2018*) is bacteriostatic and incompletely inhibits respiration, but becomes bactericidal and abolishes oxidative phosphorylation in *M. tuberculosis* lacking the alternative cytochrome BD Δ*cydAB*, *Kalia et al., 2017*; *Foo et al., 2018* or treated with a CytBD inhibitor (*Lee et al., 2021*). Although the $bcc:aa_3$ oxidase and CytBD are functionally redundant, they differ in cofactor usage for electron transfer: $bcc:aa_3$ is a heme-copper oxidase with three copper ions in the electron transport path (*Gong et al., 2018*; *Kim et al., 2015*; *Mathiyazakan et al., 2023*; *Wiseman et al., 2018*), whereas CytBD uses heme prosthetic groups but not copper (*Safarian et al., 2021*; *Figure 2A*).

To test whether diisonitrile chalkophores maintain the function of the copper-dependent $bcc:aa_3$ oxidase under copper limitation (*Figure 2A*), we generated *M. tuberculosis* Δ*nrp*Δ*cydAB*, along with control strains lacking *cydAB* alone and genetically complemented strains. We tested the effect of TTM on growth in liquid media and observed mild inhibition of growth of *M. tuberculosis* Δ*nrp* and no effect on Δ*cydAB* (*Figure 2B*). However, growth of Δ*nrp*Δ*cydAB* was severely inhibited by TTM, an effect that was reversed by genetic complementation with either *nrp* or *cydABDC* (*Figure 2B*). Assays on copper chelated agar media (bathocuproinedisulfonicacid disodium salt (BCS) or TTM) revealed a dramatic sensitization of chalkophore deficient *M. tuberculosis* by loss of the secondary oxidase with

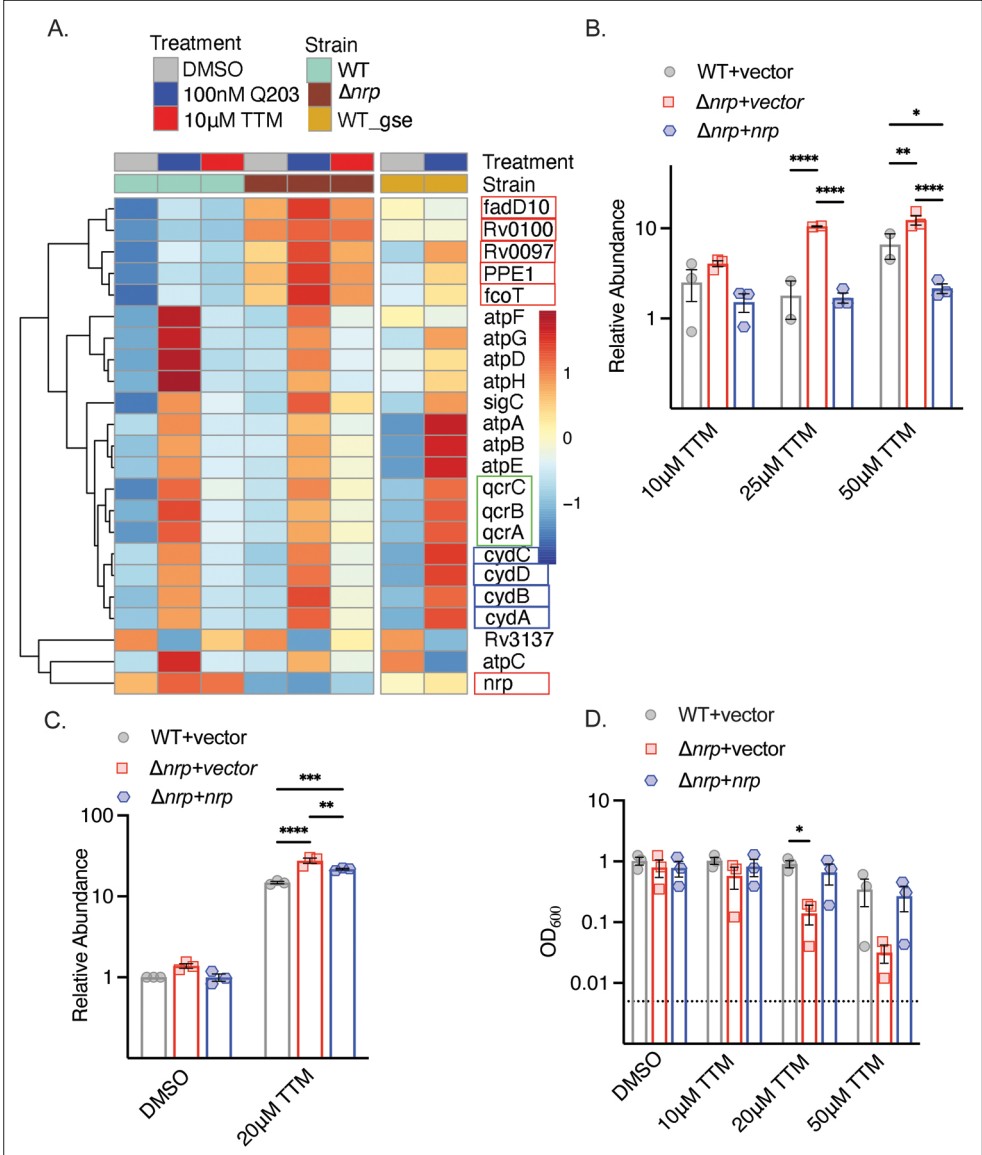

**Figure 1.** Copper deprivation in chalkophore-deficient *M. tuberculosis* mimics *bcc:aa₃* oxidase inhibition. (**A**) Heat map of transcripts encoding selected respiratory chain components determined by RNA sequencing of *M. tuberculosis* wild-type (WT) or Δ*nrp* treated with TTM or Q203. WT_GSE is the published dataset GSE159080 of *M. tuberculosis* H37Rv treated with Q203. Genes in the chalkophore cluster are boxed in red, genes encoding the cytochrome BD (CytBD) oxidase in blue, and genes encoding components of the *bcc:aa3* supercomplex are in green. (**B**) RT-qPCR of the transcript encoding CydA in *M. tuberculosis* WT, Δ*nrp*, and complemented strain treated with varying TTM concentrations for 4 hr. Error bars represent the standard error of the mean (SEM). Statistical significance determined via two-way ANOVA with Tukey correction for multiple comparisons. $*p<0.05$, $**p<0.01$, $****p<0.0001$. (**C**) RT-qPCR of the transcript encoding CydA in *M. tuberculosis* WT, Δ*nrp*, and complemented strain treated with 20 μM TTM for 24 hr. Error bars are SEM. Statistical significance determined via two-way ANOVA with Tukey correction for multiple comparisons. $**p<0.01$, $***p<0.001$, $****p<0.0001$. (**D**) Dose-dependent effect of tetrathiomolybdate (TTM) on growth of the indicated *M. tuberculosis* strains at 7 d post inoculation. The dotted line indicates the starting inoculum. Error bars are SEM. Statistical significance determined via two-way ANOVA with Tukey correction for multiple comparisons. $*p<0.05$.

The online version of this article includes the following source data for figure 1:

**Source data 1.** Raw data values for *Figure 1B–D*.

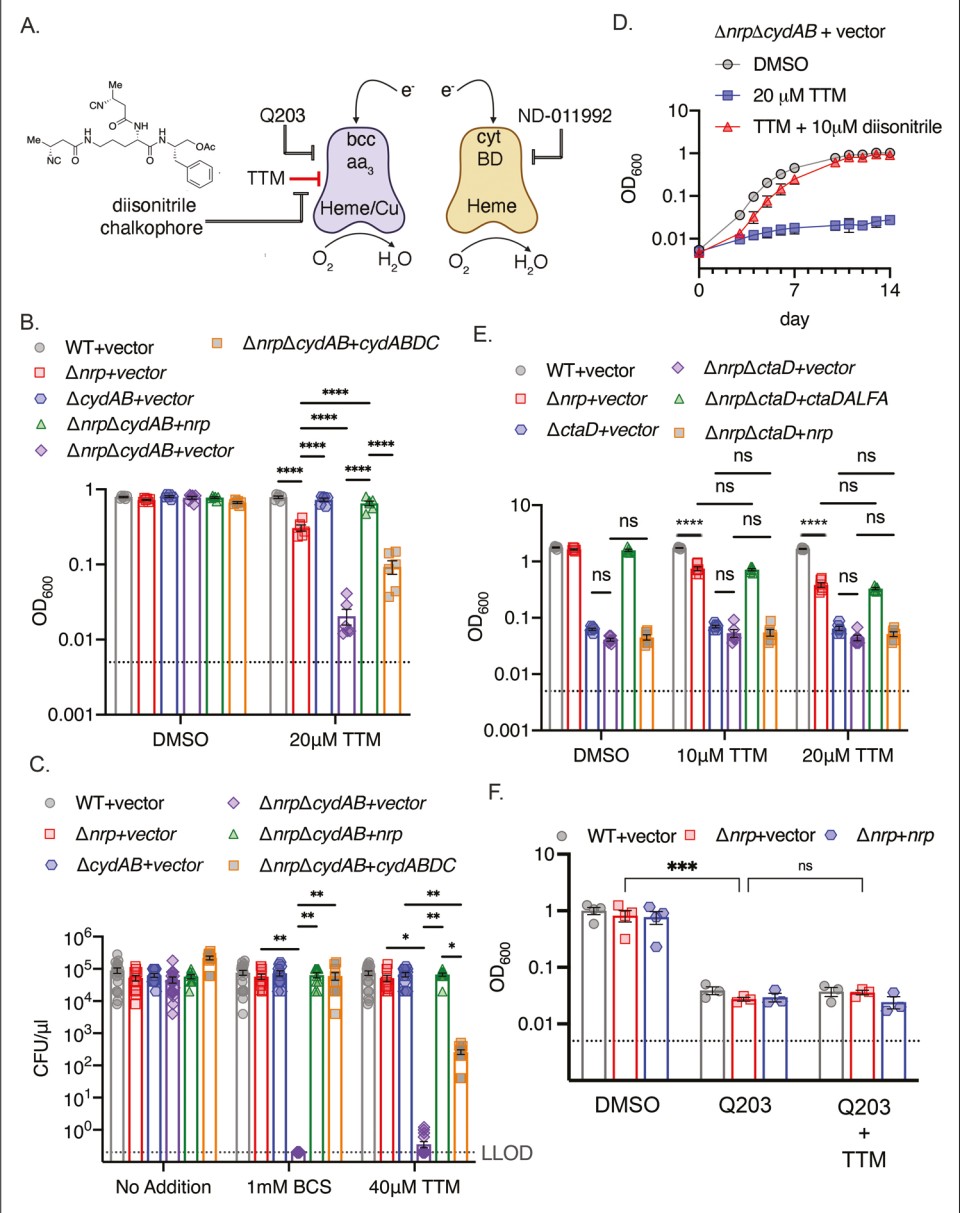

**Figure 2.** Chalkophores maintain *M. tuberculosis* viability through the heme-copper *bcc:aa₃* oxidase during copper starvation. (**A**) Schematic of the terminal respiratory oxidases of *M. tuberculosis*. The *bcc:aa₃* oxidase is a heme-copper oxidase and cytochrome BD (CytBD) is a copper-independent heme oxidase. Both transfer electrons to oxygen. Q203 is an inhibitor of *bcc:aa₃* by targeting the QcrB subunit, whereas ND-011992 targets CytBD. The two oxidases are individually dispensable due to compensation by the other oxidase, but *M. tuberculosis* lacking both is nonviable. The model to be tested is that copper chelation deprives the *bcc:aa₃* oxidase of copper and that diisonitrile chalkophores counter this copper deprivation stress. (**B**) Liquid growth assays of the indicated strains with or without 20 µM TTM treatment. OD₆₀₀ at day 10 post-inoculation displayed. Dotted line indicates starting inoculum. Error bars are SEM. Statistical significance determined via two-way ANOVA with Tukey correction for multiple comparisons. ****p<0.0001. (**C**) Bacterial survival of the indicated strains on agar media containing no addition, 1 mM BCS, or 40 µM TTM. Dotted line indicates lower limit of detection (LLOD). Error bars are SEM. Statistical significance determined via two-way ANOVA with Tukey correction for multiple comparisons. **p<0.01, *p<0.05 (**D**) The copper deprivation sensitivity of *M. tuberculosis ΔnrpΔcydAB* strain can be rescued with a synthetic diisonitrile chalkophore. Liquid growth assays of *ΔnrpΔcydAB* with DMSO, 20 µM TTM, or 20 µM TTM with 10 µM of the diisonitrile chalkophore pictured in panel A. Error bars are SEM. (**E**) The *bcc:aa₃* oxidase is the only target of copper starvation countered by diisonitrile chalkophores. Liquid growth assays of the indicated strains treated with 10 or 20 µM TTM, or DMSO vehicle control. OD₆₀₀ at day 10 post-inoculation displayed. Dotted

*Figure 2 continued on next page*

*Figure 2 continued*

line indicates starting inoculum. Error bars are SEM. Statistical significance determined via two-way ANOVA with Tukey correction for multiple comparisons. ns = not significant, ****$p<0.0001$. (**F**) The effect of copper deprivation is masked by inhibition of QcrB subunit of $bcc:aa_3$. Liquid growth assays of the indicated strains treated with Q203 (100 nM) alone or co-treated with 100 nM Q203 and 10 µM TTM. $OD_{600}$ at day 7 post-inoculation displayed. Dotted line indicates starting inoculum. Error bars are SEM. Statistical significance determined via two-way ANOVA with Tukey correction for multiple comparisons. ***$p<0.001$, ns = not significant.

The online version of this article includes the following source data and figure supplement(s) for figure 2:

**Source data 1.** Raw data for *Figure 2B–F*.

**Figure supplement 1.** Loss of *cydAB* severely sensitizes chalkophore-deficient *M. tuberculosis* to copper chelation.

**Figure supplement 1—source data 1.** Raw bacterial counts from the experiment in *Figure 2—figure supplement 1C*.

**Figure supplement 2.** Full growth curves of chalkophore deficient strains with TTM copper chelation.

**Figure supplement 2—source data 1.** Raw data showing OD600 measuremenets graphed in *Figure 2—figure supplement 2A–F*.

**Figure supplement 3.** Full growth curves in basal and copper chelation conditions of *ctaD* mutant strains.

**Figure supplement 3—source data 1.** Raw data showring OD600 values for *Figure 2—figure supplement 3A and B*.

---

6 logs of killing, a phenotype that was also complemented by *nrp* or *cydABDC* (*Figure 2C*, *Figure 2—figure supplements 1 and 2*). Complementation was incomplete with TTM compared to BCS for reasons we have not determined. To determine whether the phenotypes of Δ*nrp* are due to diisonitrile chalkophore deficiency rather than some function of the genetic element independent of the diisonitrile chalkophore itself, we first repeated the same *cydAB* synergy test with *M. tuberculosis* Δ*fadD10*, a second biosynthetic gene in the *nrp* operon that is required for chalkophore-mediated resistance to copper chelation (*Buglino et al., 2022*). *M. tuberculosis* Δ*fadD10*Δ*cydAB* was also dramatically sensitized to BCS or TTM to a similar degree as Δ*nrp*Δ*cydAB* (*Figure 2—figure supplement 1B, C*). To demonstrate that the copper deprivation sensitivity of the Δ*nrp*Δ*cydAB* strain is due to the absence of the diisonitrile lipopeptide, we synthesized an analogue of a reported *M. tuberculosis* diisonitrile chalkophore (*Mehdiratta et al., 2022*) having 4-carbon side chains (see Methods and *Supplementary file 4*). This C4 ʟ-ornithyl-ʟ-phenylalaninol acetate diisonitrile (C4-Orn-Phin-OAc) was synthesized via a route analogous to our previously reported syntheses of a *Streptomyces* diisonitrile (*Xu and Tan, 2019*). This synthetic diisonitrile efficiently rescued Δ*nrp*Δ*cydAB* from the growth inhibition imposed by TTM in liquid media (*Figure 2D*, *Figure 2—figure supplement 2*), consistent with direct mediation of this effect by the diisonitrile lipopeptide product of the *nrp* locus.

The data above indicates that diisonitrile biosynthesis is necessary to resist the effects of copper starvation and that one of the effects of this copper starvation is dysfunction of the $bcc:aa_3$ oxidase. To determine whether maintenance of respiratory oxidase function under copper starvation is the only function of diisonitrile chalkophores, we executed genetic and biochemical epistasis testing. Deleting *ctaD*, which encodes one subunit of the $bcc:aa_3$ supercomplex, alone or in diisonitrile chalkophore deficient cells (Δ*nrp*Δ*ctaD*), impaired growth of *M. tuberculosis* in copper-replete media (*Figure 2E*, *Figure 2—figure supplement 3*), but copper deprivation had no additional effect, indicating that the heme-copper oxidase complex is the only target impacted by copper starvation in these culture conditions (*Figure 2E*). Similarly, the growth inhibitory effect of copper starvation on Δ*nrp* cells was similar in degree to treatment of WT or Δ*nrp* cells with Q203, but combined treatment was not synergistic (*Figure 2F*), again indicating that the heme-copper oxidase is inactivated by copper deprivation when the diisonitrile chalkophore is missing and that no additional targets are relevant to copper deprivation-induced growth arrest in these conditions.

## Chalkophores maintain oxidative phosphorylation and ATP production

To determine whether diisonitrile chalkophores directly maintain respiration during copper deprivation, as suggested by the growth and survival data above, we measured oxygen consumption by *M. tuberculosis* using a qualitative methylene blue assay (*Kalia et al., 2017*; *Lee et al., 2021*). In this

assay, decolorization of methylene blue in a sealed tube indicates oxygen depletion, as observed in WT or $\Delta nrp\Delta cydAB$ *M. tuberculosis* treated with DMSO (*Figure 3A*), indicating intact respiration in chalkophore-deficient cells in basal conditions. Treatment with Q203 did not impair oxygen consumption in WT cells but did in $\Delta nrp\Delta cydAB$, consistent with its inhibition of the $bcc:aa_3$ oxidase (*Figure 3A*). Treatment of WT cells with Q203 in combination with ND-011992, a small-molecule inhibitor of CytBD, also blocked oxygen consumption, consistent with prior data (*Figure 3—figure supplement 1A*; *Kalia et al., 2017*; *Lee et al., 2021*). Copper chelation with 50 µM TTM had no effect on WT cells but abolished respiration in $\Delta nrp\Delta cydAB$, indicating that copper deprivation inhibits respiration through the $bcc:aa_3$ oxidase (*Figure 3A*). Similarly, $\Delta nrp$ cells grown on agar media with TTM and ND-011992 lost viability (*Figure 3—figure supplement 1C*). Taken together, these data are consistent with a model in which *M. tuberculosis* can respire via either the $bcc:aa_3$ oxidase or CytBD, with the former requiring diisonitrile chalkophore-mediated copper acquisition under copper starvation. The full effect of loss of chalkophore biosynthesis is masked by the redundancy of the oxidases, but in cells that rely only on $bcc:aa_3$ oxidase, respiration is abolished by copper deprivation.

To measure oxygen consumption more quantitatively, we adopted a spot sensor assay (*Kalia et al., 2023*), that can noninvasively measure oxygen in sealed vessels, allowing serial measurements. WT *M. tuberculosis* consumed oxygen down to the lower limit of detection, and treatment with the CytBD inhibitor ND-011992 (*Lee et al., 2021*) had no effect (*Figure 3B*). Treatment with the QcrB inhibitor Q203 delayed but did not prevent oxygen consumption, consistent with the methylene blue results and prior data (*Kalia et al., 2017*; *Lamprecht et al., 2016*; *Lee et al., 2021*). However, treatment with the combination of Q203 and ND-011992 completely inhibited oxygen consumption (*Figure 3B*). Applying this assay to diisonitrile chalkophore deficient strains, we observed that copper deprivation delayed but did not prevent respiration in $\Delta nrp$ cells (*Figure 3C*), consistent with compensation by CytBD, but abolished respiration in $\Delta nrp\Delta cydAB$ cells, a phenotype that was rescued by genetic complementation with the *nrp* gene (*Figure 3D*). To confirm that the diisonitrile chalkophore-mediated protection of respiration is accompanied by ATP depletion, we measured ATP levels in WT and chalkophore-deficient cells treated with copper deprivation or Q203. Copper deprivation with TTM or treatment with Q203 had minimal effect on WT cells (*Figure 3E*), but inhibited ATP production in the $\Delta nrp\Delta cydAB$ strain (*Figure 3E*). Taken together, these data demonstrate that diisonitrile chalkophores maintain oxidative phosphorylation by the heme-copper respiratory oxidase under copper starvation. Because the electron transport path of the $bcc:aa_3$ oxidase contains three copper sites, these findings are consistent with a model in which these copper sites become dysfunctional when copper is limiting, leading to impaired supercomplex biosynthesis or function.

In eukaryotic copper trafficking disorders in which copper becomes limiting in mitochondria, cytochrome C oxidase becomes unstable, preventing assembly (*Soma et al., 2018*). To determine if a similar mechanism is operative in mycobacterial cells, we inserted an ALFA epitope tag (*Götzke et al., 2019*) at the C terminus of CtaD in both WT and $\Delta nrp$ cells and, after confirming functionality (*Figure 2E*), examined CtaD protein levels with copper deprivation. CtaD levels remained unchanged with either TTM or BCS (*Figure 3F*, *Figure 3—figure supplement 1B*), indicating that copper deprivation is not affecting oxidase biogenesis and is more likely acting on preexisting respiratory oxidase complexes.

## Chalkophores defend oxidative phosphorylation from nutritional immunity

Respiration has emerged as an attractive target for antimycobacterials, with the ATP synthase inhibitor bedaquiline now a cornerstone of multidrug resistant (MDR) TB treatment (*Conradie et al., 2022*; *Conradie et al., 2020*) and Q203 in clinical trials (*de Jager et al., 2020*). Although it is also clear that ATP generation via oxidative phosphorylation is required for *M. tuberculosis* growth in mice (*Kalia et al., 2017*; *Lee et al., 2021*; *Beites et al., 2019*; *Cai et al., 2021*), it is less clear whether the reactive centers of the electron transport chain are compromised by the host and whether *M. tuberculosis* must defend or restore the integrity of the electron transport chain during infection. To examine this question, we infected C57BL/6 mice with *M. tuberculosis* strains lacking diisonitrile chalkophore biosynthesis, alone and in combination with deletion of the genes encoding CytBD ($\Delta nrp$ and $\Delta nrp\Delta cydAB$). The mouse attenuation phenotype of *M. tuberculosis* lacking *nrp* has been reported by several groups (*Buglino et al., 2021*; *Bhatt et al., 2018*; *Mehdiratta et al., 2022*) and indicates a mild early

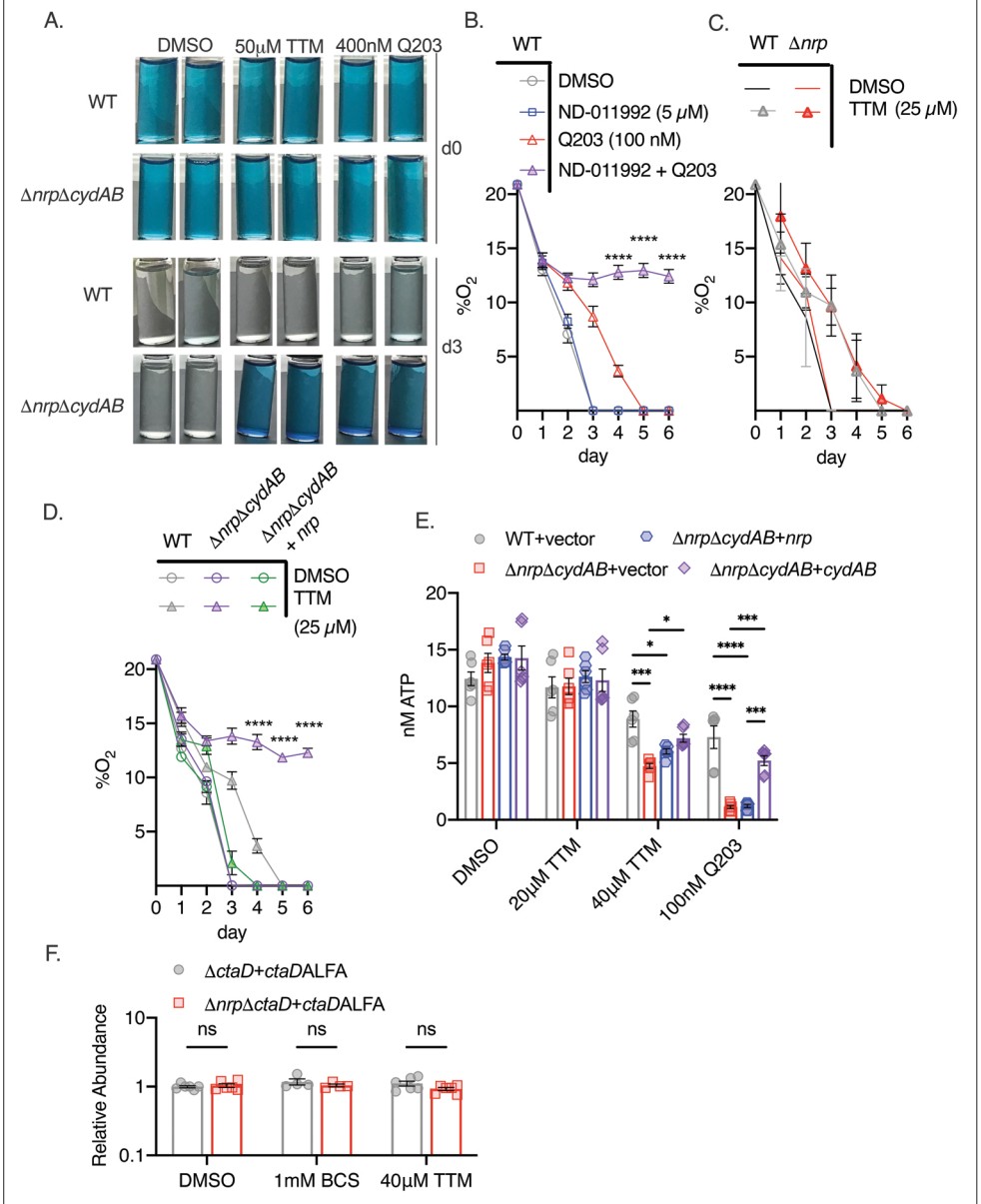

**Figure 3.** Chalkophore biosynthesis maintains oxidative phosphorylation through the heme-copper *bcc:aa₃* oxidase. (**A**) Methylene blue decolorization assay of oxygen consumption under copper deprivation (tetrathiomolybdate, TTM) or treatment with Q203 in wild-type (WT) or Δ*nrp*Δ*cydAB M. tuberculosis* at day 0 (d0) or day 3 (d3) of incubation. Clear vials indicate oxygen consumption by respiration. (**B**) Quantitative measurement of oxygen consumption using oxygen-sensitive optical sensors. WT *M. tuberculosis* treated with DMSO, ND-011992, Q203, or both ND-011992 and Q203. Oxygen measurements were taken daily. Each point represents three measurements of two biological replicates. Error bars are SEM. Statistical significance between Q203 and ND-011992 +Q203 determined via two-way ANOVA with Tukey correction for multiple comparisons. ****$p<0.0001$. (**C**) Same assay as in panel B with WT and Δ*nrp M. tuberculosis* treated with DMSO or 25 μM TTM. Error bars are SEM. (**D**) Same assay as in panel B with WT, Δ*nrp*Δ*cydAB,* or Δ*nrp*Δ*cydAB + nrp* treated with 25 μM TTM. Error bars are SEM. Statistical significance between WT and Δ*nrp*Δ*cydAB* treated with 25 μM TTM determined via two-way ANOVA with Tukey correction for multiple comparisons. ****$p<0.0001$. (**E**) Cellular ATP levels determined by BacTiter-Glo in the indicated strains treated with DMSO, 20 or 40 μM TTM, or 100 nM Q203. [ATP] determined by standard curve determined in growth media containing the same quantities of DMSO, TTM, or Q203. Error bars are SEM. Statistical significance determined via two-way ANOVA with Tukey correction for multiple comparisons. *$p<0.05$, ***$p<0.001$, ****$p<0.0001$. (**F**) Relative abundance of a CtaD-ALFA protein in *M. tuberculosis* of the indicated genotype treated with bathocuproinedisulfonic acid (BCS) or TTM. See *Figure 3—figure supplement*

*Figure 3 continued on next page*

*Figure 3 continued*

*1* for primary immunoblot data. Error bars are SEM. Statistical significance determined via two-way ANOVA with Tukey correction for multiple comparisons. ns = not significant.

The online version of this article includes the following source data and figure supplement(s) for figure 3:

**Source data 1.** Raw data for *Figure 3B–F*.

**Figure supplement 1.** Chalkophores protect the heme-copper respiratory oxidase in copper-limiting conditions.

**Figure supplement 1—source data 1.** PDF file containing original western blots for *Figure 3—figure supplement 1B*, with bands labeled.

**Figure supplement 1—source data 2.** Original uncropped western blot files for *Figure 3—figure supplement 1B*.

**Figure supplement 1—source data 3.** Bacterial counts for *Figure 3—figure supplement 1C*.

attenuation phenotype in the lungs. Loss of *cydAB* in the Δ*nrp* background dramatically exacerbated the mild attenuation of diisonitrile chalkophore-deficient strain in the lung (*Figure 4A*) and caused severe attenuation in the spleen (*Figure 4B*). Complementation with the *nrp* gene restored virulence to WT levels, demonstrating that chalkophore-mediated protection of the respiratory chain is a critical virulence function of *M. tuberculosis*.

To investigate the host pressure that targets the respiratory chain, we hypothesized that neutrophils, which express several metal binding proteins such as calprotectin, might be relevant. However, efficient depletion of neutrophils by administration of a Ly6G antibody (*Figure 4—figure supplement 1A*) did not reverse the severe attenuation of the Δ*nrp*Δ*cydAB* strain in the lungs or spleen (*Figure 4C and D*). Similarly, infection of SCID mice, which lack adaptive immunity, also had no effect on diisonitrile chalkophore deficient *M. tuberculosis* titers in lung or spleen (*Figure 4E and F*) and the rapid mortality of SCID mice infected with WT *M. tuberculosis* was not evident in SCID mice infected with Δ*nrp*Δ*cydAB* (*Figure 4—figure supplement 1B*), indicating that non-adaptive immunity is fully capable of controlling respiratory chain compromised *M. tuberculosis*.

## Discussion

We have identified diisonitrile chalkophore biosynthesis as an *M. tuberculosis* virulence mechanism that defends against copper starvation during infection. Although it is well established that phagosomal pathogens experience both metal excess and metal deprivation in the host, the general paradigm of copper and zinc nutritional immunity is that high phagosomal levels of these metals are deployed as antimicrobial effectors to limit pathogen growth (*Shi et al., 2014*; *Festa et al., 2011*; *Botella et al., 2011*; *Boudehen et al., 2022*; *Darwin, 2015*). Our data indicate that *M. tuberculosis* must also cope with copper deprivation as a host-inflicted stress during infection and that *M. tuberculosis* deploys diisonitrile chalkophores to acquire copper in the host. Diisonitrile chalkophores display an extremely high affinity for copper ions (*Wang et al., 2017*; *Xu and Tan, 2019*; *Chen et al., 2024*), and their ability to overcome copper deprivation by strong copper chelators in vitro and host immunity in vivo indicates that they perform an analogous function to bacterial siderophores, which scavenge iron from host iron limitation. Given the abundant data indicating a high copper environment in the macrophage phagosome, the sites and circumstances of the copper-deprived niche of *M. tuberculosis* remain to be determined, but could include distinct subcellular compartments (i.e. phagosomal vs cytosolic), phases of infection (ie early infection vs post-adaptive immunity), or bacterial localization within different lung compartments such as cavities.

Although deprivation of metals from pathogens, including iron and zinc, is a well-recognized mechanism of innate immunity, in most cases, the specific bacterial targets that ultimately mediate the antimicrobial function of metal deprivation are not clearly defined and are assumed to be pleiotropic due to the numerous essential metal-dependent enzymes in the bacterial cell. However, our data indicate that, in culture, the copper deprivation countered by the diisonitrile chalkophore system targets a single membrane enzyme complex, the *bcc:aa*$_3$ supercomplex, a heme-copper respiratory oxidase. *M. tuberculosis* deploys two respiratory oxidases, *bcc:aa*$_3$ and CytBD, which are redundant for bacterial viability. This dual oxidase arrangement provides respiratory chain flexibility during chemical inhibition of each oxidase, and some evidence suggests that respiratory chain flexibility promotes host

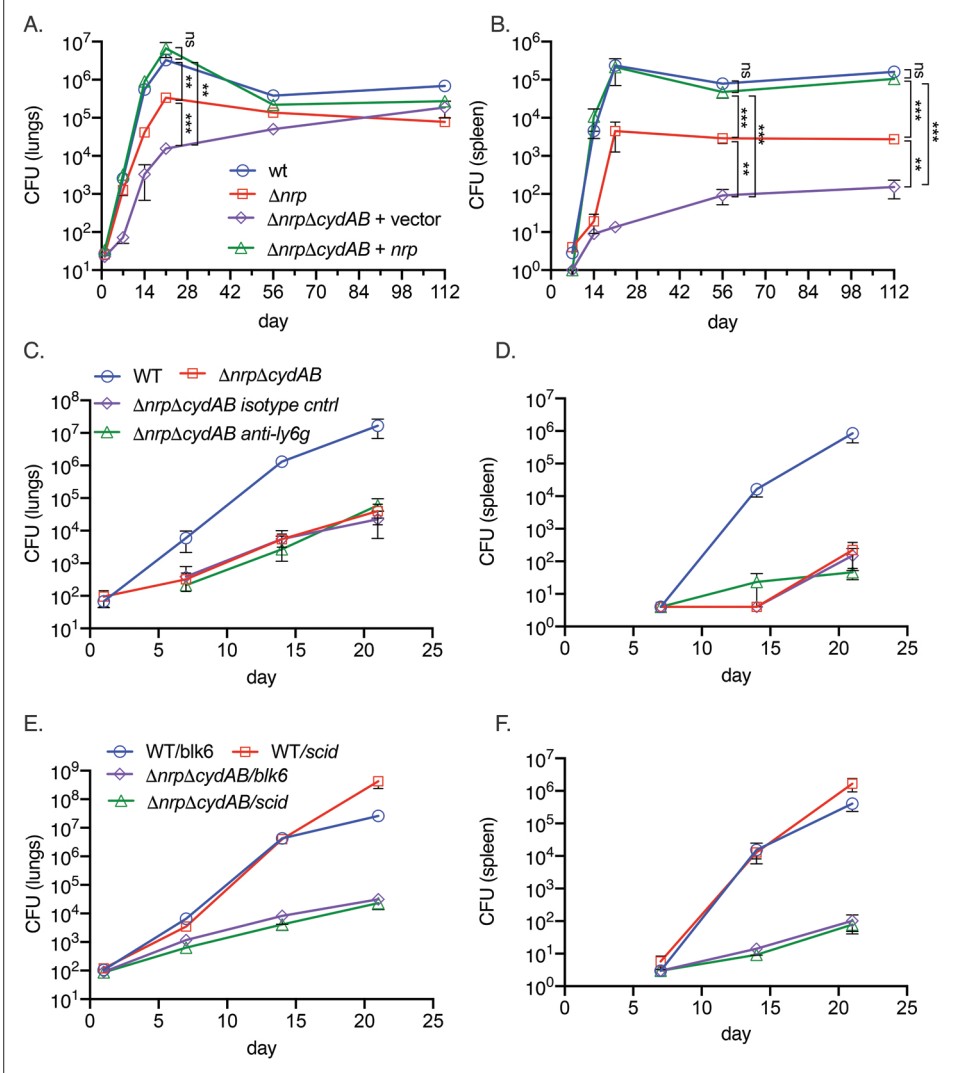

**Figure 4.** Respiratory chain flexibility is critical for *M. tuberculosis* virulence. (**A, B**) Bacterial titers in the lung (**A**) or spleen (**B**) in mice infected with *M. tuberculosis* wild-type (WT), Δ*nrp*, Δ*nrp*Δ*cydAB,* or Δ*nrp*Δ*cydAB* + *nrp*. Error bars are SEM. Statistical significance determined via two-way ANOVA with Tukey correction for multiple comparisons. Not significant (ns), **$p<0.01$, and ***$p<0.001$. (**C, D**) Copper deprivation by the host is independent of neutrophils. Bacterial titers in the lung (**C**) or spleen (**D**) in mice infected with *M. tuberculosis* WT or Δ*nrp*Δ*cydAB,* or Δ*nrp*Δ*cydAB* + *nrp* treated with isotype control antibodies or anti-Ly6G antibodies to deplete neutrophils. Flow cytometric quantitation of neutrophil depletion is provided in Figure S5. Error bars are SEM. (**E, F**) Copper deprivation by the host is independent of adaptive immunity. Bacterial titers in the lung (**E**) or spleen (**F**) in C57BL/6 J or C57BL/6 SCID mice infected with *M. tuberculosis* WT or Δ*nrp*Δ*cydAB*. Error bars are SEM.

The online version of this article includes the following source data and figure supplement(s) for figure 4:

**Source data 1.** Bacterial counts from mouse organs plotted in *Figure 4A–F*.

**Figure supplement 1.** Attenuation of chalkophore-deficient *M. tuberculosis* is independent of neutrophils and adaptive immunity.

**Figure supplement 1—source data 1.** Raw data for *Figure 4—figure supplement 1A and B*.

adaptation (*Kalia et al., 2017*; *Beites et al., 2019*; *Cai et al., 2021*). The copper centers of respiratory oxidases that participate in electron flow to oxygen are conserved from bacteria to mitochondria and represent a membrane-exposed electroreactive center susceptible to damage. We demonstrate that the chalkophore is critical to maintaining the function of the *bcc:aa₃* supercomplex, both in vitro and in vivo. However, our data does not indicate whether the chalkophore:Cu complex is an intrinsic part of the supercomplex assembly system or simply replenishes the copper pools of the mycobacterial

cell. Although chalkophore-mediated protection of the *bcc:aa₃* supercomplex is an important virulence function, we cannot exclude the possibility that additional copper-dependent enzymes use chalkophore-delivered copper during infection. Our data also reveal that *M. tuberculosis* deploys multilayered strategies to maintain oxidative phosphorylation in the face of host immune pressure. *M. tuberculosis* deploys a backup oxidase, CytBD, which can support *M. tuberculosis* virulence when the *bcc:aa₃* oxidase is dysfunctional, such as when QcrB is inhibited by Q203. However, the role of this backup oxidase in pathogenesis was unclear. Although CytBD is induced in mouse lung, peaking at 21 d post-infection (*Shi et al., 2005*), and is required for optimal fitness of *M. tuberculosis* in the lung in competition experiments (*Cai et al., 2021*), *M. tuberculosis* lacking CytBD is fully virulent (*Kalia et al., 2017*). Our results reveal the essential role of this compensatory oxidase in virulence, which is only evident when the *bcc:aa₃* oxidase is inactivated in vivo in *M. tuberculosis* lacking chalkophore biosynthesis, thereby revealing a complex system for maintaining respiration. These studies further strengthen the rationale for targeting CytBD for antibiotic development (*Safarian et al., 2021*; *Harikishore et al., 2023*).

The respiratory chain of *M. tuberculosis* has emerged as a promising drug target. Our data indicates that diisonitrile chalkophore biosynthesis, as a mechanism of protection for the respiratory chain during infection, may provide an alternative approach to target this critical energy-generating system. Beyond its importance as a drug target, this study identifies a new mechanism of resistance to copper deprivation that, in concert with CytBD, is part of a central virulence strategy of *M. tuberculosis* to protect oxidative phosphorylation during infection.

## Materials and methods

### Reagents

Middlebrook 7H10 Agar, 7H9 Broth, dextrose, Tween-80, bovine serum albumin (BSA), and UltraPure DNase/RNase-free distilled water were purchased from Fisher Scientific. Ammonium tetrathiomolybdate (TTM), bathocuproinedisulfonic acid, disodium salt (BCS), dimethyl sulfoxide (DMSO), copper sulfate, zinc sulfate, magnesium sulfate, calcium chloride, and ATP were purchased from Millipore Sigma. Biotechnology (BT) grade Chelex 100 resin, sodium form, was purchased from BioRad. Q203 (Telacebec) was purchased from AbMole. ND-011992 was synthesized as previously described (*Lee et al., 2021*). The diisonitrile lipopeptide analogue was synthesized and characterized as described below.

### General growth conditions, strains, and DNA manipulations

*M. tuberculosis* Erdman WT and mutant strains were grown and maintained in 7H9 media (broth), or on 7H10 (agar) supplemented with 10% OADC (oleic acid, albumin, dextrose, saline), 0.05% glycerol (7H9-OADC/7H10-OADC). Broth cultures were additionally supplemented with 0.02% Tween-80. Chromosomal deletion mutations were generated by specialized transduction utilizing the temperature-sensitive phage phAE87. Mutant strains were confirmed by PCR followed by sequencing. For complete strain list with relevant features, see *Supplementary file 1*. Plasmids utilized in this study were generated using standard molecular techniques and are listed with their features in *Supplementary file 2*.

### Growth assays

For liquid growth assays, the indicated strains were pre-grown in non-chelated media until reaching an $OD_{600}$ value of ~1.0. Cells were then collected by centrifugation (3700 × *g*, 10 min) and washed twice with Chelexed phosphate buffered saline with 0.02% Tween-80 (PBS Tween-80). Growth assays were initiated at a calculated $OD_{600}$ of 0.005 by the addition of 1 mL of washed culture at an $OD_{600}$ of 0.05–9 mL of replete 7H9-ADS (albumin, dextrose, saline) generated as described previously (*Buglino et al., 2022*). TTM, BCS, Q203, ND-011992, and/or synthetic diisonitrile chalkophore were then added at the indicated concentration. Growth at 37 °C was assayed via daily $OD_{600}$ measurements.

For agar growth assays, cells were pre-grown and washed as indicated above. Washed cells were then normalized to an $OD_{600}$ of 0.1, serially diluted from $10^0$-$10^{-6}$ in PBS Tween-80, and spotted on 7H10-OADC plates containing the indicated concentrations of TTM, BCS, ND-011992, or DMSO

vehicle control in triplicate. Plates incubated at 37 °C, 5% $CO_2$ were imaged after 14 d of incubation to show relative growth, and CFU counts were enumerated after 21 d of incubation.

## RNA preparation

Triplicate 25 mL cultures of strains indicated above were grown in 7H9-ADS until they reached an $OD_{600}$ of 0.3–0.5. Cultures were then treated with the indicated concentration of TTM, 100 nM Q203, or a DMSO vehicle control, for 5 hr. Following a duplicate wash with Chelex-treated PBS Tween-80, cells were collected by centrifugation (3,700×g, 10 min), suspended, and resuspended in 1 mL of TRIzol reagent. Cells in 1 mL of TRIzol were mechanically disrupted with zirconia beads via 2×45 s pulses in a BioSpec Mini24 BeadBeater with 5 min intervening rest periods on ice. Following lysis, beads were removed by centrifugation (20,000×g, 5 min) and total RNA was isolated using the Direct-zol RNA miniprep kit (Zymo Research) as directed by the manufacturer.

## RT-qPCR

DNAse treatment of RNA purified as noted above was performed using the Turbo DNA-free kit (Invitrogen). A total of 500 ng of resulting total RNA was used to synthesize cDNA via random priming utilizing the iScript cDNA Synthesis Kit (Biorad). Real-time qPCR was performed on QuantStudio 6 Pro (Applied Biosystems) using iTaq Universal SYBR Green Supermix (Biorad). For *cydA* expression, normalized cycle threshold (CT) was determined relative to the housekeeping gene *sigA*. Reactions without reverse transcriptase were included for each sample and excluded DNA contamination as a source of amplification signal. Relative expression level was calculated using the formula $2^{-(CT_{cydA}-CT_{sigA})}$. Sequences of gene-specific primers are as follows: For *cydA*, 5'-GTCATCGAAGTGCCCTATGT-3' and 5'- CTGGTATTCCTGCTGCAGAT-3', and for *sigA*, 5'- CGTCTTCATCCCAGACGAAAT-3' and 5'-CGAC GAAGACCACGAAGAC-3'.

## RNA sequencing and data analysis

After RiboGreen quantification and quality control by Agilent BioAnalyzer, 500 ng of total RNA underwent ribosomal depletion with the NEBNext rRNA Depletion Kit (Bacteria) (NEB catalog # E7850) and library preparation with the TruSeq Stranded Total RNA LT Kit (Illumina catalog # RS-122–1202) according to instructions provided by the manufacturer with 8 cycles of PCR. Samples were barcoded and run on a NovaSeq 6000 in a PE100 run, using the NovaSeq 6000 S4 Reagent Kit (200 Cycles) (Illumina). On average, 27 million paired reads were generated per sample. Post-run demultiplexing and adapter removal were performed and fastq files were inspected using FastQC (*Andrews, 2010*). Trimmed fastq files were then aligned to the reference genome (*M. tuberculosis* H37Rv; NC_000962.3) using bwa mem (*Li et al., 2009*). BAM files were sorted and merged using samtools (*Li and Durbin, 2009*) and gene counts were obtained using featureCounts from the Bioconductor Rsubread package (*Liao et al., 2014*). Differentially expressed genes were identified using the DESeq2 R package (*Love et al., 2014*) and subsequent analysis of gene expression was performed as described (*Kolde, 2025*; *R Development Core Team, 2013*). RNA sequencing data is deposited in the SRA under accession number BioProject: PRJNA1249518.

## Immunoblotting

Duplicate 20 mL cultures of chelated 7H9-ADS replete for all ions were inoculated at an $OD_{600}$ of 0.02. Upon reaching an $OD_{600}$ of 0.6–0.7, cultures were treated with the indicated concentration of TTM, or DMSO vehicle control, for 24 hr. Cells were collected by centrifugation (3700×g, 10 min) washed once with 1 mL of lysis buffer (350 mM sodium chloride, 20 mM Tris pH 8.0, 1 mM 2-mercaptoethanol) prior to suspension in 0.8 mL of lysis buffer plus ~100 µL of zirconia beads. Lysis was performed by 3×45 s pulses in a BioSpec Mini24 beadbeater with 5 min intervening rest periods on ice. Beads and debris were removed by centrifugation at 20,000×g for 15 min at 4 °C, the resulting supernatant was mixed 1:1 with 2 x Laemmli sample buffer supplemented with 0.1 M dithiothreitol (DTT). 20 µL of each sample, heated for 10 min at 100 °C, was then separated on 4–12% NuPAGE Bis-Tris polyacrylamide gels. Separated proteins were transferred to nitrocellulose and probed with the indicated antibodies. Antibodies used in this study are monoclonal anti-*E. coli* RNA-polymerase β (BioLegend), sdAb anti-ALFA tag-HRP (NanoTag Biotechnologies). Chemiluminescence was visualized using SuperSignalWest Pico Plus chemiluminescent substrate (Thermo Scientific). Blots were imaged on an iBright FL1000

imager (Thermo Fisher Scientific). CtaD ALFA blots were quantitated using ImageJ software. Relative ALFA signal was normalized to corresponding Rpoβ loading control levels.

## Methylene blue assay

Cultures were grown in 7H9-ADS to an $OD_{600}$ of 0.5–0.7, washed twice in Chelex-treated PBS Tween-80, and adjusted to an $OD_{600}$ of 0.15 with fresh 7H9-ADS and incubated with indicated concentrations of TTM, Q203, ND-011992, or DMSO for 4 hr. Following incubation, 4 ml screw-cap glass vials were filled with the cultures and methylene blue dye was added at a final concentration of 0.001%. All vials were sealed tightly using a PTFE/rubber seal (Thermo Scientific) and incubated at 37 °C in an anaerobic container (Benton Dickinson GasPak EZ container systems) for 3 d.

## Oxygen consumption measurements

Oxygen consumption was assessed using PSt6 sensor spots and a Fibox 4 trace instrument (PreSens Precision Sensing GmbH Am BioPark, Germany) (*Kalia et al., 2023*). Screw cap glass vials (Thermo Scientific) containing the sensor spot PSt6 were filled with cultures grown in 7H9-ADS at a calculated $OD_{600}$ of 0.005 and the indicated concentrations of TTM, Q203, ND-011992, or DMSO. The vials were tightly sealed using a PTFE/rubber seal (Thermo Scientific) and incubated at 37 °C in an anaerobic pouch (Benton Dickinson GasPak EZ pouch systems). Percent oxygen was measured daily via the Fibox 4 trace oxygen meter without unsealing the anaerobic environment.

## ATP quantitation

ATP levels were quantified with the BacTiter-Glo (Promega). The indicated strains were first pre-grown to an $OD_{600}$ of 0.5–1.0 in 7H9-OADC. Cells were then washed twice with PBS Tween-80 and suspended in 7H9-OADC at an $OD_{600}$ of 0.05. 96-Well plates were inoculated with 100 µL of washed culture in replicate wells. The indicated concentrations of TTM, Q203, or DMSO vehicle control were then added, and plates were incubated at 37 °C, 0.05% $CO_2$ for 24 hr. 100 µL of BacTiter-Glo was then added to each well and plates were incubated 15 additional minutes at 37 °C, 5% $CO_2$ prior to reading on a SpectraMax M3 plate reader. [ATP] in test samples was quantified against standard curves of ATP.

## Synthesis of diisonitrile chalkophores

Details of synthesis and characterization are provided in *Supplementary file 4*.

## Aerosol infection of mice

8–10 wk-old C57BL/6 J (JAX stock number 00064) and 8–10 wk B6.Cg-*Prkdc$^{scid}$/SzJ* (stock number 001913, RRID:IMSR_JAX:001913) were purchased from The Jackson Laboratory. All purchased mice were rested within our animal facility to normalize microbiota for 2 wk. Care, housing, and experimentation on laboratory mice were performed in accordance with the National Institute of Health guidelines, and the approval of the Memorial Sloan Kettering Institutional Animal Care and Use Committee (IACUC). Strains for infection were grown to an $OD_{600}$ of 0.5–0.7 in 7H9-OADC, washed twice with PBS Tween-80, followed by brief sonication to disrupt aggregates. Final inoculums were prepared by suspending $8 \times 10^7$ CFU in 10 mL of sterile water. Mice were exposed to $4 \times 10^7$ CFU in a Glas-Col aerosol exposure unit, a dose calibrated to deliver 10–50 CFU per mouse. At the indicated time points post-infection, five individual mice were humanely euthanized and both lungs and spleens were harvested for CFU determination. Organ homogenates were cultured on 7H10-OADC plates, and CFU was enumerated after 28 d of incubation at 37 °C, 5% $CO_2$. Twenty additional SCID mice (10: WT *M. tuberculosis*, 10: Δ*nrp*Δ*cydAB*) infected as described above, were used to assess survival. Infected mice were monitored daily and humanely euthanized upon frank signs of morbidity.

## Neutrophil depletion

Mice were treated intraperitoneally with anti-Ly6G (BioXcell, clone 1A8, RRID:AB_1107721) or isotype control (BioXcell, clone 2A3, RRID:AB_1107769) every 48 hrs. Lungs were harvested aseptically and single cell suspensions prepared by homogenization in a Bullet Blender with 2.0 mM zirconium oxide beads for 3 min followed by 100 micron filtration. After lysis of red blood cells with Gibco ACK lysing buffer and blocking of Fc receptors (Invitrogen CD16/CD32 clone 93, RRID:AB_467133), cells were stained with live/dead aqua (Thermo) and antibodies to CD45 (FITC, clone 30-F11, BD Biosciences,

RRID:AB_394609), CD11b (BUV661, clone M1/70, BD Biosciences), CD11c (APC-R700, clone N418, RRID:AB_2744277), Ly6C (BV605, clone AL-21, BD Biosciences, AB_2737949), CD80 (PerCP Cy5.5, clone 16–10 A1 AB_1727514), CD86 (Pacific Blue, clone GL-1, BioLegend, RRID:AB_493466), MHC II (BUV395, clone 2G9, BD Biosciences RRID:AB_2741827), GR1 (PE, clone RB6-8C5, BD Biosciences, RRID:AB_398532), and Ly6G (APC, clone 1A8, BD Biosciences, RRID:AB_394208) and analyzed by flow cytometry on a BD Fortessa cytometer after paraformaldehyde fixation.

## Acknowledgements

This work was supported by R01AI138446 and P30 CA008748. We acknowledge the use of the Integrated Genomics Operation Core, funded by the NCI Cancer Center Support Grant (CCSG, P30 CA08748), Cycle for Survival, and the Marie-Josée and Henry R Kravis Center for Molecular Oncology. We thank Christina Stallings for advice about neutrophil depletion and George Sukenick and Rui Wang (MSK) for expert NMR and mass spectral support.

## Additional information

### Competing interests

Michael S Glickman: Declares equity and consulting fees from Vedanta biosciences and consulting fees from Fimbrion therapeutics. The other authors declare that no competing interests exist.

### Funding

| Funder | Grant reference number | Author |
|---|---|---|
| National Institute of Allergy and Infectious Diseases | R01AI138446 | John A Buglino<br>Yaprak Ozakman<br>Anna Benjamin |
| National Cancer Institute | P30 CA008748 | John A Buglino<br>Yaprak Ozakman<br>Chad E Hatch<br>Anna Benjamin<br>Derek S Tan |

The funders had no role in study design, data collection and interpretation, or the decision to submit the work for publication.

### Author contributions

John A Buglino, Yaprak Ozakman, Conceptualization, Investigation, Methodology, Writing – review and editing; Chad E Hatch, Anna Benjamin, Investigation; Derek S Tan, Conceptualization, Supervision; Michael S Glickman, Conceptualization, Supervision, Investigation, Writing – original draft, Writing – review and editing

### Author ORCIDs

John A Buglino  https://orcid.org/0000-0001-8961-8672
Yaprak Ozakman  https://orcid.org/0000-0003-3777-0546
Chad E Hatch  https://orcid.org/0000-0001-6808-5117
Anna Benjamin  https://orcid.org/0000-0001-6159-6440
Derek S Tan  https://orcid.org/0000-0002-7956-9659
Michael S Glickman  https://orcid.org/0000-0001-7918-5164

### Ethics

Care, housing, and experimentation on laboratory mice were performed in accordance with the National Institute of Heath guidelines, and the approval of the Memorial Sloan Kettering Institutional Animal Care and Use Committee (IACUC), protocol 01-11-030.

Reviewer #1 (Public review): https://doi.org/10.7554/eLife.105794.3.sa1
Reviewer #2 (Public review): https://doi.org/10.7554/eLife.105794.3.sa2

Reviewer #3 (Public review): https://doi.org/10.7554/eLife.105794.3.sa3
Author response https://doi.org/10.7554/eLife.105794.3.sa4

## Additional files

### Supplementary files

Supplementary file 1. Strains used in this study.

Supplementary file 2. Plasmids used in this study.

Supplementary file 3. RNA sequencing dataset.

Supplementary file 4. Chemical synthesis methods, $^{1}$H NMR, and $^{13}$C NMR spectra.

1MDAR checklist

### Data availability

RNA sequencing data is deposited in the SRA under accession number BioProject: PRJNA1249518. Source data for all figures is included.

The following dataset was generated:

| Author(s) | Year | Dataset title | Dataset URL | Database and Identifier |
|---|---|---|---|---|
| Buglino J, Ozakman Y, Bean J, Glickman MS | 2025 | Chalkophore mediated respiratory oxidase flexibility controls *M. tuberculosis* virulence | https://www.ncbi.nlm.nih.gov/bioproject/PRJNA1249518 | NCBI BioProject, PRJNA1249518 |

The following previously published dataset was used:

| Author(s) | Year | Dataset title | Dataset URL | Database and Identifier |
|---|---|---|---|---|
| Hasenoehrl EJ, Berney M | 2020 | Transcriptional response of *Mycobacterium tuberculosis* to inhibition of Cytochrome bd oxidase and Cytochrome bcc:aa3 oxidase | https://www.ncbi.nlm.nih.gov/geo/query/acc.cgi?acc=GSE159080 | NCBI Gene Expression Omnibus, GSE159080 |

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

# Appendix 1

**Appendix 1—key resources table**

| Reagent type (species) or resource | Designation | Source or reference | Identifiers | Additional information |
|---|---|---|---|---|
| Gene (*M. tuberculosis*) | *nrp* | | erdman_0118 | |
| Gene (*M. tuberculosis*) | *cydA* | | erdman_1783 | |
| Gene (*M. tuberculosis*) | *cydB* | | erdman_1782 | |
| Gene (*M. tuberculosis*) | *cydD* | | erdman_1781 | |
| Gene (*M. tuberculosis*) | *cydC* | | erdman_1780 | |
| Gene (*M. tuberculosis*) | *ctaD* | | erdman_3330 | |
| Gene (*M. tuberculosis*) | *fadD10* | | erdman_0116 | |
| Strain (*Escherichia coli*) | DH5α | Lab Stock | ATCC SCC2197 | Plasmid Maintenance Strain |
| Strain (*Escherichia coli*) | EL350/pHAE87 | Lab Stock | | Phage packaging strain |
| Strain (*Mus musculus*) Female | B6.Cg-Prkdc(scid)/SzJ | Jackson Laboratory | Stock No 001913 | |
| Strain (*Mus musculus*) Female | c56bl 6 | Jackson Laboratory | Stock No 000664 | |
| Strain (*M. tuberculosis*) | M.tb Erdman (WT, EG2) | Lab Stock | ATCC 35801 | Animal Passaged |
| genetic reagent (*M. tuberculosis*) nrp KO | *nrp::hygR* | buglino et al. 2022 | Δ*nrp* | Chromosomal Deletion of nt 21–7515 of Edman_0118 by double crossover recombination |
| Genetic reagent (*M. tuberculosis*) cydAB KO | *cydAB::hygR* | This Study | Δ*cydAB* | Chromosomal deletion of nt1 of erdman_1783 to post erdman_1782 stop codon by double crossover recombinaiton |
| Genetic reagent (*M. tuberculosis*) fadD10 KO | *fadD10::hygR* | This Study | Δ*fadD10* | Chromosomal deletin of nt 85–1552 of erdman_0116 by double crossover recombination |
| Genetic reagent (*M. tuberculosis*) ctaD KO | *ctaD::hygR* | This Study | Δ*ctaD* | Chromosomal deletion of nt 115–1723 of erdman_3330 by double crossover recombination |
| Genetic reagent (*M. tuberculosis*) nrp cydAB KO | *nrp::loxP cydAB::hygR* | This Study | Δ*nrp*Δ*cydAB* | |
| Genetic reagent (*M. tuberculosis*) fadD10 cydAB KO | fadD10::loxP cydAB::hygR | This Study | ΔfadD10ΔcydAB | |
| Genetic reagent (*M. tuberculosis*) nrp ctaD KO | nrp::loxP ctaD::hygR | This Study | ΔnrpΔctaD | |

*Appendix 1 Continued on next page*

*Appendix 1 Continued*

| Reagent type (species) or resource | Designation | Source or reference | Identifiers | Additional information |
|---|---|---|---|---|
| Transfected construct (*M. tuberculosis*) vector | pMV306 | Lab Stock | +vector | L5 attP intergrating mycobacterium plasmid |
| Transfected construct (*M. tuberculosis*) nrp | pJAB823 | buglino et al. 2022 | +*nrp* | L5 attP integrated: AA 1–2512 erdman_0118 HSP60 promoter |
| Transfected construct (*M. tuberculosis*) cydABDC | pJAB900 | This Study | +*cydABDC* | L5 attP integrated: 330 bp 5' of erdman_1783 start to stop codon of erdman_1780 |
| Transfected construct (*M. tuberculosis*) ctaD | pJAB863 | This Study | +*ctaD* ALFA | L5 attP integrated: 310 bp 5' of erdman_3330 start to stop condon ALFA tag inserted between nt 5881 and 5882 |
| Antibody (anti-ALFA) | sdAb anti-ALFA | NanoTag Biotechnologies | N1505-HRP | 1:4000 dilution |
| Antibody (anti-RpoB) | Anti-*E. coli* RNA pol B | Biolegend | 663903 | 1:10000 dilution |
| Antibody (anti-mouse HRP) | IgG (H+L) Goat anti-Mouse, HRP | Fisher Scientific | 626520 | 1:10000 dilution |
| Antibody (anti-ly6G) | InVivoMAb anti-mouse Ly6G | Bio-X-Cell | BE0075-1-25MG | |
| Antibody (isotype control) | InVivoMAb rat IgG2a isotype control, anti-trinitrophenol | Bio-X-Cell | BE0089-25MG | |
| Recombinant DNA reagent | | | | |
| Sequence-based reagent (qPCR primer) sigA | oSigA-1 | Intergrated DNA Technologies | | 5'-cgtcttcatcccagacgaaat-3' |
| Sequence-based reagent (qPCR primer) sigA | oSigA-2 | Intergrated DNA Technologies | | 5'-cgacgaagaccacgaagac-3' |
| Sequence-based reagent (qPCR primer) cydA | *cydA* FWD set3 | Intergrated DNA Technologies | | 5'-gtcatcgaagtgccctatgt-3' |
| Sequence-based reagent (qPCR primer) cydA | *cydA* REV set3 | Intergrated DNA Technologies | | 5'-ctggtattcctgctgcagat-3' |
| Peptide, recombinant protein | | | | |
| Commercial assay or kit | In-Fusion Snap Assembly Master Mix | Takara Bio USA | 638949 | |
| Commercial assay or kit | NEBNext rRNA Depletion Kit | NEB | E785OS | |
| Commercial assay or kit | TruSeq Stranded Total RNA kit | Illumina | 20020599 | |
| Commercial assay or kit | Novaseq 6000 S4 Reagent Kit | Illumina | 20028313 | |
| Commercial assay or kit | TURBO DNA-free kit | Fisher Scientific | AM1907 | |

*Appendix 1 Continued on next page*

*Appendix 1 Continued*

| Reagent type (species) or resource | Designation | Source or reference | Identifiers | Additional information |
|---|---|---|---|---|
| Commercial assay or kit | Phusion High Fidelity Polymerase | Fisher Scientific | F530L | |
| Commercial assay or kit | GeneJET Plasmid Miniprep Kit | Fisher Scientific | FERK0503 | |
| Commercial assay or kit | Zymo Research Corporation Direct-zol RNA MiniPrep | Fisher Scientific | 50-444-622 | |
| Commercial assay or kit | Taq Universal SYBR Green Supermix | BioRad | 1725122 | |
| Commercial assay or kit | iScript cDNA Synthesis Kit | BioRad | 1708891 | |
| Commercial assay or kit | GeneJet Gel Extraction Kit | Fisher Scientific | FERK0692 | |
| Chemical compound, drug | methylene blue | Fisher Scientific | S25429 | |
| Chemical compound, drug | dimethyl sulfoxide (DMSO) | Sigma | D2650 | |
| Chemical compound, drug | Ammonium tetrathiomolybdate (TTM) | Sigma | 323446 | |
| Chemical compound, drug | Bathocuproinedisulfonic acid disodium salt (BCS) | Sigma | B1125 | |
| Chemical compound, drug | Telacebec (Q203) | AbMole BioScience | M5297 | |
| Chemical compound, drug | ND-011992 | This Study | | synthesized by Tan lab, see methods |
| Chemical compound, drug | diisonitrile | This Study See supplemental methods | | |
| Chemical compound, drug | TRIzol Reagent | Fisher Scientific | 15-596-026 | |
| Software, algorithm | fastqc | | http://www.bioinformatics.babraham.ac.uk/projects/fastqc | |
| Software, algorithm | bwa mem | *Li and Durbin, 2009* | | |
| Software, algorithm | samtools | *Li et al., 2009* | | |
| Software, algorithm | Bioconductor Rsubread package | *Liao et al., 2014* | | |
| Software, algorithm | DESeq2 R package | *Love et al., 2014* | | |
| Software, algorithm | R | The R Project for Statistical Computing | https://www.R-project.org | |
| Other | NUPAGE 4–12% BT Gel | Fisher Scientific | NPO312BOX/NPO336BOX | |
| Other | Protran Nitrocellulose Hybridization Transfer Membrane | Perkin Elmer | NBA08C001EA | |
| Other | Fibox 4 trace instrument | PreSens | https://www.presens.de/products/detail/fibox-4-trace | |
| Other | Oxygen detection sensor spot PSt6 | PreSens | https://www.presens.de/products/detail/oxygen-sensor-spot-sp-pst6-nau | |

