## [Editor Report · eLife Assessment]

In this **important** study, the authors advance our understanding of copper uptake by chalkophores and their targeted metalloproteins in *Mycobacterium tuberculosis*. These **convincing** data demonstrate that chalkophore-acquired copper is solely incorporated into the Mtb bcc:aa3 copper-iron respiratory oxidase under low copper conditions, and that chalkophore-mediated protection of the respiratory chain is critical to Mtb virulence. These findings may be leveraged for drug discovery and will be of broad interest to those studying bacterial pathogenesis.

---

## [Referee Report · Reviewer #1 (Public review)]

Summary:

It is known that the nrp operon is induced by copper deprivation and encodes the synthesis of chalkophores. The authors carried out a genetic analysis that revealed transcriptional differences for WT and Mtb∆nrp when exposed to the copper chelator tetrathiomolybdate (TTM). The authors found that copper chelation results in upregulation of genes in the chalkophore cluster as well as genes involved in the respiratory chain: including, components of the heme-dependent oxidase CytBD and subunits of the bcc:aa3 heme-copper oxidase. Utilizing several knockout variants and inhibitors, the authors showed that copper starvation survival requires chalkophore synthesis and that copper starvation results in dysfunctional bcc:aa3 oxidase. By monitoring oxygen consumption, they go on to show that copper deprivation inhibits respiration through the bcc:aa3 oxidase. Lastly, the authors compare virulence of WT Mtb, Mtb∆nrp and MtbΔnrpΔcydAB strains in mice spleen and lung. The Mtb∆nrp strain showed mild attenuation, but virulence in MtbΔnrpΔcydAB was severely attenuated and complementation with the chalkophore biosynthetic pathway restored Mtb virulence. These results suggest that chalkophore mediated protection of the respiratory chain is critical to Mtb virulence, and that redundant respiratory oxidases within Mtb provide respiratory chain flexibility that may promote host adaptation.

This new information about Mtb biology may be leveraged for drug discovery, highlighting that the Mtb respiratory pathway is a promising drug target, where one may target the Mtb chalkophore biosynthetic pathway in conjunction with CytBD, to obliterate Mtb.

Strengths: Overall, the paper is very clear and well written, with thorough and well-thought-out experimentation.

No weaknesses.

Comments on revisions:

The authors have addressed all the reviewers' comments.

---

## [Referee Report · Reviewer #2 (Public review)]

Summary:

This is a well-written manuscript that clearly demonstrates that the nrp encoded diisonitrile chalkophore is necessary for function of the bcc-aa3 oxidase supercomplex under low copper conditions. In addition, the study demonstrates the chlakophore is important early during infection when copper sequestration is employed by the host as a method of nutritional immunity.

Strengths:

The authors use genetic approaches, including single and double mutants of chalkophore biosynthesis, and both the Mtb oxidases. Use a copper chelators to restrict copper in vitro. A strength of the work was the use of a synthesized a Mtb chalkophore analogue to show chemical complementation of the mutant nrp locus. Oxphos metabolic activity was measured by oxygen consumption and ATP levels. Importantly, the study demonstrated that chalkophore, especially in a strain lacking the secondary oxidase, was necessary for early infection and ruled out a role for adaptive immunity in the chalkophore lacking Mtb by use of SCID mice. It is interesting that after two weeks of infection and onset of adaptive immunity the chalkophore is not required, which is consistent with the host environment switching from a copper restricted to copper overload in phagosomes.

Weaknesses:

None noted

---

## [Referee Report · Reviewer #3 (Public review)]

Summary:

In this manuscript, the group of Glickman expand on their previous studies on the function of chalkophores during growth of and infection by *Mycobacterium tuberculosis*. Previously, the group had shown that chalkophores, which are metallophores specific for the scavenging of copper, are induced by *M. tuberculosis* under copper deprivation conditions. Here, they show that chalkophores, under copper limiting conditions, are essential for the uptake of copper and maturation of a terminal oxidase, the heme-copper oxidase, cytochrome bcc:aa3. As *M. tuberculosis* has two redundant terminal oxidases, growth of and infection by *M. tuberculosis* is only moderated if both the chalkophores and the second terminal oxidase, cytochrome bd, are inhibited.

Strengths:

A strength of this work is that the lab-culture experiments are complemented with mice infection models, providing strong indications that host-inflicted copper deprivation is a condition that *M. tuberculosis* has adapted to for virulence.

Weaknesses:

Because the phenotype of *M. tuberculosis* lacking chalkophores is similar, if not identical, to using Q203, an inhibitor of cytochrome bcc:aa3, the authors propose that the copper-containing cytochrome bcc:aa3 is the only recipient of copper-uptake by chalkophores. A minor weakness of the work is that this latter conclusion is not verified under infection conditions and other copper-enzymes might still be functionally required during one or more stages of infection.

Comments on revisions:

I thank the authors for carefully addressing my suggestion to the original submission and congratulate them on their work.

---

## [Author Response]

The following is the authors’ response to the original reviews

**Response to public reviews:**

We thank the reviewers for their careful evaluation of our manuscript and appreciate the suggestions for improvement. We will outline our planned revisions in response to these reviews.

**Reviewer 2**: “The one exception is the claim that "maintenance of respiration is the only cellular target of chalkophore mediated copper acquisition." While under the in vitro conditions tested this does appear to be the case; however, it can't be ruled out that the chalkophore is important in other situations. In particular, for maintenance of the periplasmic superoxide dismutase, SodC, which is the other *M. tuberculosis* enzyme known to require copper.”And**Reviewer 3**: “Because the phenotype of *M. tuberculosis* lacking chalkophores is similar, if not identical, to using Q203, an inhibitor of cytochrome bcc:aa3, the authors propose that the coppercontaining cytochrome bcc:aa3 is the only recipient of copper-uptake by chalkophores. A minor weakness of the work is that this latter conclusion is not verified under infection conditions and other copper-enzymes might still be functionally required during one or more stages of infection.

Both comments concern the question of whether the *bcc*:*aa3* respiratory oxidase supercomplex is the only target of chalkophore delivered copper. In culture, our experiments suggest that *bcc*:*aa3* is the only target. The evidence for this claim is in Figure 2E and F. In 2E, we show that *M. tuberculosis* D_ctaD_ (a subunit of *bcc*:*aa3*) is growth impaired, copper chelation with TTM does not exacerbate that growth defect, and that a D_ctaD_D_nrp_ double mutant is no more sensitive to TTM than D_ctaD_. These data indicate that role of the chalkophore in protecting against copper deprivation is absent when the *bcc*:*aa3* oxidase is missing. Similar results were obtained with Q203 (Figure 2F). Q203 or TTM arrest growth of *M. tuberculosis* D_nrp,_ but the combination has no additional effect, indicating that when Q203 is inhibiting the *bcc*:*aa3* oxidase, the chalkophore has no additional role. However, we agree with the reviewers that we cannot exclude the possibility that during infection, there is an additional target of chalkophore mediated Cu acquisition. We have added this caveat to the discussion of revised version of this manuscript.

**Response to Reviewers Recommendations for the authors:**

**Reviewing Editor Comments:**
In addition to the specific recommendations below, there was consensus that the conclusions/discussion should contextualize that the results cannot exclude that in other conditions (such as in infection), enzymes other than cytochrome bcc:aa3 receive copper from the chalkophore system.
**Reviewer #1 (Recommendations for the authors):**
(1) In the introduction, the authors mention that the nrp operon is only present in pathogenic Mtb and Mycobacterium marinum but not non-pathogenic mycobacterium. Is the nrp operon present in other pathogenic mycobacterium such as in M. leprae, M. avium or M. abscessus?

Bhatt et al (PMID 30381350) presented an analysis of the distribution of *nrp* gene clusters in mycobacteria and concluded that *M. bovis*, *M. leprae* and *M. canetti* clearly encode *nrp* genes. *M. marinum* has been shown to have a functional chalkophore biosynthetic cluster, but the presence of this system in other mycobacteria awaits experimental validation. We have added the Bhatt reference to this sentence in the introduction.

(2) Figure 1A - it would be helpful if the genes were grouped and labeled as per their purpose (for example, CytBD components, bcc:aa3 components). While these are described in the text, the genes belonging to the chalkophore cluster are not defined in the text, and are thus not easily identified in the figure.

The order of genes in the heatmap is determined by unsupervised clustering as indicated by the dendrogram to the left of the heatmap. To highlight chalkophore and CytBD genes, we have added color coding to the gene names and explained this color coding in the legend.

(3) Figure 2B/2C - it is interesting that complementation of ΔnrpΔcydAB with cydABCD does not rescue growth to Δnrp levels. Is there an explanation for this?AND(4) Figure 2C - BCS is not introduced in the text for this figure nor are the results described - which seems like an oversight. It is interesting that BCS treatment does have a full rescue with cydABCD complementation, while TTM treatment does not. Is there an explanation for this?

We thank the reviewer for raising this issue. We have attempted several different complementation constructs, including CydAB alone and different promoters, to address the partial complementation in question. However, we do not have an adequate explanation for this partial complementation. As the reviewer notes, the partial complementation is only evident with TTM, not BCS. However, we cannot speculate on the reason for this difference at present. We have added a note to the text in the results section noting this difference.

(5) Figure 2F - is there a reason for the change in TTM concentrations (50 μM TTM vs 10 μM TTM)? Is the concentration for Q203 in both single treatment and combinatory tests 100nM?

We have clarified the 100nm Q203 concentration in the figure legend. To avoid confusion, we have removed the 50µM TTM condition from panel F because the growth inhibition phenotype of 10µM is shown in panel E and is the comparator for the combined TTM/Q203 condition in panel F.

(6) Figure 3A - I assume d0 = day 0, d3 = day 3. This should be defined.

We have modified the legend to clarify these abbreviations.

(7) Figure 4B - as complementation of nrp for ΔnrpΔcydAB returns levels back to WT, I assume there is no attenuation with ΔcydAB alone? Clarification would be appreciated.

The mouse phenotype of *M. tuberculosis* D_cydAB_ is reported here:

https://www.pnas.org/doi/10.1073/pnas.1706139114#sec-1 and this paper is reference 22 of the paper and was noted in the discussion.

**Reviewer #2 (Recommendations for the authors):**

In vitro conditions that require SodC could reveal a role for the chalkophore (ie., exposure to extracellular or periplasmic superoxide stress under low iron conditions). Some minor confusion exists with the terminology around the two oxidases found in *M. tuberculosis*. The bcc:aa3 oxidase is a supercomplex between the reductase and oxidase complexes. This point should be clarified in the introduction as the term supercomplex isn't used until later in line 194 and without definition. Referring to the bcc:aa3 supercomplex as an oxidase is fine but is sometimes confusing especially when mentioning the target of Q203 is the oxidase as it targets the reductase portion of the supercomplex.

We thank the reviewer for this point. We have modified the text to refer to the supercomplex at first mention and modified subsequent mentions to be clearer.

In the RNA preparation section boxes appear in several places where spaces should be.

We do not see these boxes so we suspect this is a conversion error of some type.

**Reviewer #3 (Recommendations for the authors):**
The authors have very carefully performed their studies and their main conclusions are amply supported by the data. The manuscript is also very clearly written, and easily accessible to a broad audience interested in both bioinorganic chemistry and mycobacteria. I have two recommendations:

(1) I agree that the evidence shows that chalkophores provide copper to cytochrome bcc:aa3. Under lab-culture conditions, it could well be that, when cytochrome bd is deleted or inhibited, cytochrome bcc:aa3 is rate limiting. Under lab-culture conditions, it is also clear that only the expression of a select number of enzymes is affected. However, this does not mean that cytochrome bcc:aa3 is the ONLY enzyme that receives copper from chalkophores. Thus, under infection conditions, other copper enzymes might be important. For instance, *M. tuberculosis* expresses a Cu-Zn superoxide dismutase. In summary, perhaps the authors would consider changing the wording of statements such as that in Figure 2E and the conclusions drawn in the discussion.

This comment concerns the question of whether the *bcc*:*aa3* respiratory supercomplex is the only target of chalkophore delivered copper. In culture, our experiments suggest that the supercomplex is the only target. The evidence for this claim is in Figure 2E and F. In 2E, we show that *M. tuberculosis* D_ctaD_ (a subunit of the *bcc*:*aa3* supercomplex) is growth impaired, copper chelation with TTM does not exacerbate that growth defect, and that a D_ctaD_D_nrp_ double mutant is no more sensitive to TTM than D_ctaD_. These data indicate that role of the chalkophore in protecting against copper deprivation is absent when the *bcc*:*aa3* supercomplex is missing. Similar results were obtained with Q203 (Figure 2F). Q203 or TTM arrest growth of *M. tuberculosis* D_nrp,_ but the combination has no additional effect, indicating that when Q203 is inhibiting *bcc*:*aa3*, the chalkophore has no additional role. However, we agree with the reviewers that we cannot exclude the possibility that during infection, there is an additional target of chalkophore mediated Cu acquisition. We have added the following to the discussion: “Although chalkophore mediated protection of the *bcc*:*aa3* supercomplex is an important virulence function, we cannot exclude the possibility that additional copper dependent enzymes use chalkophore delivered copper during infection.”

(2) There is a difference between copper-uptake (e.g. by chalkophores) and the maturation of metallo-enzymes. A short paragraph discussing knowledge from other bacteria in this area would help understand the role chalkophores (e.g. see 10.1128/mBio.00065-18 or 10.1111/mmi.14701). This could possibly be extended with a genome analysis to check which other proteins are present in *M. tuberculosis*.

We thank the reviewer for this point. We agree that our data does not distinguish between (1) a generic role for the chalkophore in copper uptake, with the ultimate candidate metalloenzyme rendered dysfunctional by copper loss, and (2) the chalkophore being an intrinsic part of the cytochrome maturation pathway and interacting directly with the target enzymes. We have added this point to the discussion but have not otherwise added the suggested full discussion of metalloenzyme maturation as we believe this discussion is beyond the scope of our data.

Finally, can I suggest the labels d0 and d3 are made clearer in Figure 3A (and defined in the legend).

We have modified the legend to be clearer.